# Development of High-Performance Hydrogen-Air Fuel Cell with Flourine-Free Sulfonated Co-Polynaphthoyleneimide Membrane [note 1]

**DOI:** 10.3390/membranes13050485

**Published:** 2023-04-29

**Authors:** Ulyana M. Zavorotnaya, Igor I. Ponomarev, Yulia A. Volkova, Vitaly V. Sinitsyn

**Affiliations:** 1A.M. Prokhorov Institute of General Physics RAS, Scientific Center of Materials and Technologies, Vavilova St. 38, 119991 Moscow, Russia; ulyanamzav@mail.ru; 2Physics Faculty, National Research University High School of Economics, Myasnitskaya 20, 101000 Moscow, Russia; 3A.N. Nesmeyanov Institute of Organoelement Compounds, Vavilova St. 28, GSP-1, 119991 Moscow, Russia; gagapon@ineos.ac.ru (I.I.P.); yvolk@ineos.ac.ru (Y.A.V.); 4Institute of Solid State Physics RAS, 2 Academician Ossipyan Str., 142432 Chernogolovka, Russia

**Keywords:** sulfonated co-polynaphtoyleneimide membrane, hydrophilic/hydrophobic block, fluorine-free proton exchange membrane, Nafion, PEMFC, membrane-electrode assembly fabrication, current–voltage characteristic, optimal operating temperature, MEA power output

## Abstract

This paper presents research on the technological development of hydrogen-air fuel cells with high output power characteristics using fluorine-free co-polynaphtoyleneimide (co-PNIS) membranes. It is found that the optimal operating temperature of a fuel cell based on a co-PNIS membrane with the hydrophilic/hydrophobic blocks = 70/30 composition is in the range of 60–65 °C. The maximum output power of a membrane-electrode assembly (MEA), created according to the developed technology, is 535 mW/cm^2^, and the working power (at the cell voltage of 0.6 V) is 415 mW/cm^2^. A comparison with similar characteristics of MEAs based on a commercial Nafion 212 membrane shows that the values of operating performance are almost the same, and the maximum MEA output power of a fluorine-free membrane is only ~20% lower. It was concluded that the developed technology allows one to create competitive fuel cells based on a fluorine-free, cost-effective co-polynaphthoyleneimide membrane.

## 1. Introduction

The energy facilities based on hydrogen-air fuel cells with proton exchange membranes (PEMFC) are of great interest for modern energetic development. The thermodynamics limit of such generator efficiency is ~83%, but in practice, it is currently around 60–65% [1,2]. Nevertheless, PEMFC energy capacity is higher (with the same mass dimensions) than that of lithium-ion batteries, which are the most widely and actively used in various energetic applications at present. The refueling time of PEMFC power units is an order of magnitude shorter than that of lithium-ion batteries, which makes these systems very attractive for transport applications. The existing renewable energy sources are intermittent and unstable and hence undermine the stable operation of power systems, thus opening temporal and spatial gaps between the consumption of energy by consumers and its availability [3]. PEMFC’s electricity generation does not depend on external conditions, which makes it very promising to create hybrid installations based on a renewable system with the possibility of “green” hydrogen production and hydrogen-air fuel cells. Moreover, a working PEMFC only releases electricity, heat, and water, which fully corresponds to “green energy” and contributes to reducing carbon dioxide emissions [4,5,6,7,8,9].

At the present time, a proton exchange membrane based on perfluorinated sulfonic acid Nafion^®^ is the most commonly applied separation membrane in PEMFC. This membrane has been developed in the 1960s by DuPont [10]. Lately, short-chain polymeric derivatives of the Nafion membrane, such as Aquivion, Flemion, and Nyflon, have become increasingly widespread [11,12]. However, this type of membrane has a number of significant disadvantages that limit the wider distribution of hydrogen-air fuel cells:Non-ecological and environmental pollution dangers due to fluorine compound utilization when preparing the membrane and disposing of it [13,14];The restriction of an operating temperature range due to the reduction in transport and mechanical characteristics at a temperature above 90 °C, which is associated with a sharp decrease in the polymer’s water content [15,16];A significant decrease in the proton conductivity at low humidity (<50%), which is associated with the poor water-retaining capacity of perfluorinated sulfopolymer [13,14,15,16];Accelerated membrane dehydration and a sharp increase in gas permeability at PEMFC operation in the temperature range from 90 to 120 °C [15,17,18,19,20,21,22];The noncompetitive high cost of membrane fabrication, which is associated with both the technological difficulties of polymer synthesis and its restricted production due to the presence of fluorine in the chemical structure [6,23,24].

The following strategies were applied to solve these problems: the modification of a perfluorosulfonated ionomer structure by means of composite inorganic doping and the development of fluorine-free sulfonated hydrocarbon polymers [25,26,27,28]. The last of these strategies is successful and today a large number of hydrocarbon proton exchange membranes that meet all the requirements for their practical application have already been developed [29,30,31]. Moreover, their transport characteristics are comparable to or even higher than those of fluorinated analogs. The fluorine-free hydrocarbon polymer synthesis and the proton exchange membrane fabrication based on them are much simpler and cheaper due to the absence of fluorine in their chemical structure, and they are easy to dispose of.

Earlier, it was shown that hydrocarbon polymers of the polynaphthoyleneimide class (further co-PNIS) were notable because of their high transport characteristics [32,33]. The production of proton-exchanged membranes from sulfonated polynaphtoyleneimides (SPNI) based on industrial 1,4,5,8-naphthalene tetracarboxylic acid (NTDA) anhydride with six-membered imide rings was carried out in a number of works [34,35,36]. However, a distinctive feature of the membranes studied in this work that distinguishes them from other membranes of the polynaphthoylenimide class is the ODAS/MDAC combination of monomers, which allows us to improve the overall hydrolytic stability while maintaining high proton conductivity and mechanical strength. The chemical structure of a co-PNIS membrane consists of hydrophilic 4,4′-diaminodiphenyl oxide-2,2′-disulfoacid (ODAS) blocks and hydrophobic (rather less hydrophilic) 6,6-bis-methylenedianthranylic acid (MDAC) blocks [35,36]. It was found that the membrane transport and mechanical characteristics could be changed by the variation of ODAS/MDAC blocks [33,34]. In addition, a special zirconium cross-link between ODAS and MDAC blocks allowed us to fabricate self-moistening co-PNIS membranes, which are very useful for PEMFC [33,36]. The performed studies indicate that the co-polynaphthoyleneimide membrane with a composition ratio of ODAS/MDAC = 70/30 (hereinafter co-PNIS_70/30_) is optimal in terms of mechanical strength, proton conductivity, water self-diffusion coefficient compared with other compositions membrane of this type [33]. The ratio of sulfonated and non-sulfonated fragments chosen in this work (70/30% mol.) guarantees high proton conductivity of the membranes, acceptable strength properties, and dimension stability of the material for the MEA operation. The presence of free carboxyl groups in the polymer chain improves the solubility of the copolymer in DMSO (green solvent), causes increased intermolecular interactions, and catalyzes the recycling reaction of the naphthoylenimide cycle in the case of hydrolytic processes occurring in an aqueous medium, protecting the polymer from destruction during the MEA operation. The prospects of their utilization in methanol fuel cells were demonstrated in [35].

This work aimed to develop a technology for the fabrication of a membrane-electrode assembly (MEA) using a co-PNIS_70/30_ membrane that possessed high output power performance comparable to the power characteristics of the MEA on a commercial Nafion membrane.

## 2. Materials and Methods

### 2.1. Co-PNIS with ODAS/MDAC = 70/30 Synthesis

DMSO, phenol, triethylamine, 4,4′-diaminodiphenyl ether, benzoic acid, sulfuric acid, oleum (65% SO_3_), and sodium hydroxide were purchased from Acrus. 6,6-bis-methylenedianthranylic acid (MDAC) (Vitas-M Laboratory, Champaign, IL, USA) and 1,4,5,8-napthalenetetracarboxylic acid dianhydride (NTDA) (Sigma Aldrich, Burlington, MA, USA) were used without further purification. 4,4′-Diaminodiphenyl oxide-2,2′-disulfoacid (ODAS) was synthesized as described in [37]. For the preparation of co-PNIS polymer films, the chemical components ODAS (0.5044 g, 0.0014 M), MDAC (0.1718 g, 0.0006 M), NTDA (0.5364 g, 0.002 M), benzoic acid catalyst (0.34 g, 0.0028 M), triethylamine solubilizer of ODAS (0.303 g, ~0.003 M), and phenol or DMSO as solvents (10.0 g) were placed in a three-piece flask equipped with a stirrer and a capillary for introducing argon [35,36]. The reaction mass mixture was heated in an argon flow to 80 °C while stirring until all monomers were dissolved. Then, the temperature was increased, and the reacting mixture was kept for 24 h at 120–140 °C. Then, it was cooled down to 100 °C and diluted with phenol or DMSO containing 0.12 g of Zr acetylacetonate (~10 wt% on PNIS). Zirconium was used for the polymerization of the monomers, which is illustrated in Figure 1b. A useful feature of Zr is its affinity for water, which makes the proton exchange membrane self-moistening and creates additional hydration centers. As a result, the membrane dries more slowly when placed in an anhydrous atmosphere and its transport characteristics do not decrease for a longer time period.

To cast the membranes, the reaction solutions were filtered and then poured onto a glass substrate for drying at 60 °C. The films were removed from the glass and additionally kept for 2 h in vacuum at 150 °C to eliminate residual solvent. The chemical structure of the co-PNIS_70/30_ polymer under study is shown in Figure 1. The membrane composition used in the work consists of ODAS/MDAC = 70/30.

It is well known that the transport characteristics of proton exchange membranes strongly depend on their sulfonation degree [38,39,40]. To determine the degree of sulfonation of the polymer, the IEC parameter (ion exchange capacity) is used, which is proportional to the content of sulfonic groups in the polymer. Initially, the IEC theoretical value was determined using the concentrations of the chemical components for co-PNIS synthesis. Then, the experimental IEC parameter value was determined by titration. Samples for titration were pre-weighed on an analytical balance, then placed in a glass with distilled water and kept for 1 h at (80 ± 3) °C. After that, the samples were transferred into flasks with 20 mL of 1 M NaCl solution and kept for 30 min under normal conditions. Then, 2 drops of an indicator (phenolsulfophthalein C_19_H_14_O_5_S) were added to each flask and titrated with 0.01 M NaOH solution.

The ion exchange capacity value was calculated using the following equation:(1)$$IEC=\frac{0.001\times V_{NaOH}\times C_{NaOH}}{m},$$
where *V_NaOH_* is the volume of the alkali solution, mL; *C_NaOH_* is the concentration of the alkali solution, mol/L; and m is the dry membrane weight, g. We found that the IEC of the co-PNIS_70/30_ membrane was 2.13 meq/g.

### 2.2. Fabrication of Membrane-Electrode Assemblies and Investigations of Their Current–Voltage Characteristics

A membrane-electrode assembly (MEA) included a proton exchange membrane with two gas diffusion electrodes placed on both sides of the membrane. This electrode is a gas diffusion layer (GDL) with a deposited catalyst that provides a three-phase contact among the reagent gas, the ionic membrane, and electronic (GDL) conductors. The gas diffusion electrodes could be attached or hot pressured to the membrane during MEA fabrication. Both methods have been tested during the development of MEA fabrication using a co-polynaphthoyleneimide membrane. The screen-printing method was applied for catalyst deposition onto the GDL surface, which made it possible to obtain a uniform active layer and reliably control the Pt loading. To create MEA, a Pt/C catalyst with a platinum content of 48.98% was used. The catalyst loading was 0.2 mg/cm^2^ on the anode electrode and 0.4 mg/cm^2^ on the cathode electrode. The difference in the Pt loading for the cathode and the anode allows one to reduce the MEA cost without a significant decrease in its power characteristics because the main reaction occurs at the cathode electrode, and, accordingly, a somewhat smaller amount of platinum at the anode can be used.

In the case of the MEA fabrication that involves attaching the gas diffusion electrodes to the membrane on both sides, the whole construction was contained in the measurement cell. During the MEA fabrication process involving hot-pressing, the membrane and two gas diffusion electrode assemblies were initially placed between metal sheets with a Teflon gasket of the appropriate thickness to avoid membrane deformation. Hot-pressing was carried out on an isostatic press (Carver M 3853) at a temperature of 130 °C and a pressure of 8 MPa for 3 min. These parameters were optimized at the development of the MEA fabrication containing a Nafion-type membrane.

Figure 2 schematically demonstrates the standard MEA fabrication technology used at the initial stage of this work (Section 3.1). In this work, the most optimal parameters for the MEAs fabrication based on the co-PNIS_70/30_ membrane were determined, which allowed one to develop a high-performance fuel cell. The results of these studies are given in Section 3.2.

All investigations were carried out on industrial MEAs with an active area of 18 cm^2^, which is usually used in the assembly of 30–300 watt batteries. During electrochemical studies, the MEA was placed into a measuring cell with a heating element and two independent channels for supplying fuel and air. The MEAs’ electrochemical characteristics were measured using the Arbin test facility at hydrogen and airflow rates of 0.4 L/min and 2.5 L/min, respectively. The experiments were carried out at 100% humidity of fuel and air without back pressure. The MEA break-in procedure includes exposure in the potentiostatic mode (0.6 V) until stationary values of the current density were reached [41,42,43,44]. The cyclic tests in the voltage range from *U*_OCV_ to 0.1 V (where *U*_OCV_ is the open circuit potential) with a sweep rate of 5 mV/s were carried out to determine the MEA current–voltage characteristics. The studies were carried out using the Elins P-40X potentiostat as an electronic load.

## 3. Results and Discussion

### 3.1. Characteristics of MEAs Fabricated by Standard Technology with the Co-Polynaphthoyleneimide Membrane

Figure 3 shows *I*–*U* and *I*–*P* characteristics after the activation of MEAs (break-in procedure) obtained on the co-PNIS_70/30_ membrane with attached and pressed electrodes. The standard hot-pressing technology developed for the MEA with a Nafion membrane (see Section 2.2) was applied at this stage of the studies. The characteristics of MEAs with hot-pressing electrodes attached to the Nafion 212 membrane using the same method are also shown in Figure 3 for comparison.

The open circuit potential for both MEAs based on a co-PNIS_70/30_ membrane (with attached and pressed GDLs) was U_OCV_ ≈ 1 V, which indicated some deviation from the theoretical value (1.23 V). The deviation may be due to hydrogen crossover through the membrane or associated with the kinetics peculiarities of the equilibration of the cathode potential [45]. It should be noted that the drop of the MEA voltage from U_OCV_ (activation losses) in both cases was ~0.14 V at the activation current of ~5 mA/cm^2^, which is insignificant and acceptable for a high-performance fuel cell [35]. The maximum and operating (at an output voltage of 0.6 V) power values were 169 mW/cm^2^ and 112 mW/cm^2^ for the MEA with attached electrodes, and 365 mW/cm^2^ and 309 mW/cm^2^ for the MEA with pressed electrodes.

As can be seen in Figure 2, the maximum and operating (at an output cell voltage of 0.6 V) power characteristics of MEAs with a Nafion 212 membrane are significantly higher than those of MEAs with a co-PNIS_70/30_ membrane and are equal to 748 mW/cm^2^ and 465 mW/cm^2^, respectively. One of the factors affecting the difference in the power values is a higher level of ohmic losses (the *I*–*U* linear part slope) in fuel cells based on co-polynaphthoyleneimide membranes compared to those in a fuel cell with a Nafion 212 membrane. Moreover, the ohmic losses for the MEA with a co-PNIS_70/30_ membrane fabricated using attached electrodes are significantly higher than those for the MEA with a co-PNIS_70/30_ membrane fabricated using pressed electrodes (Figure 2). This fact is a direct consequence of the deterioration of the interfacial contacts between the electrolyte and electrodes, which obviously should be improved with further development of the technology.

On the other hand, as can be seen from the *I*–*U* curves presented in Figure 2, the current density of MEAs based on co-polynaphthoyleneimide membranes increases with a decrease in voltage from 0.9 V to 0.4 V, and then almost does not change and vertically drops, which indicates the existence of a limiting factor. It can be assumed that these *I*–*U* behaviors caused by a GDL (H23C3, Fraudenberg) were used for the standard MEA fabrication that was developed for fuel cell operation under low humidity conditions and, correspondently, for the low moisture absorption of the membrane. It is known that under full hydration conditions, aromatic hydrocarbon membranes are characterized by higher water content than Nafion [38,39,40,41,42,43,44,45,46,47]. In this regard, the electrodes of MEAs based on co-PNIS_70/30_ membranes are more susceptible to flooding than those of MEAs based on the perfluorinated analogs. Perhaps, when using another type of GDL (for example, H23CX653, Freudenberg, Weinheim, Germany (see the following website for detailed specifications: https://www.fuelcellstore.com/freudenberg-carbon-paper-h23cx653, accessed on 3 April 2023) which is specially prepared for the fuel cell operations under high humidity conditions (high moisture absorption of the membrane), the electrode flooding effect will be essentially suppressed. In H23CX653 GDLs, water removal is more efficient than in the case of H23C3 GDLs (https://www.fuelcellstore.com/freudenberg-h2315-i2-c3, accessed on 3 April 2023) due to the channel system peculiarities in carbon materials resulting in the flooding reduction. Therefore, it was decided to compare the characteristics of MEAs with the two above-mentioned GDL types during further development of a high-performance fuel cell based on a poly-naphthoylenimide membrane. However, the use of the H23CX653 GDL requires additional studies on the determination of optimal hot-pressing parameters, and the results of these studies are presented below.

### 3.2. Development of Technology for the Fabrication of a High-Performance MEA with a Co-Polynaphthoyleneimide Membrane

The solution to the most obvious problem causing low power characteristics due to electrode flooding in an MEA with a co-polynaphtoyleneimide membrane was the initial process of testing the technology development of high-performance fuel cell fabrication. For this purpose, the MEA based on a co-polynaphtoyleneimide membrane with a GDL (H23CX653, Freudenberg, Weinheim, Germany) was prepared and studied under the standard parameters of the hot-pressing procedure (see Section 2.2). Figure 4 presents the current–voltage and power characteristics of the MEA fabricated on the basis of a co-PNIS_70/30_ membrane with H23C3 and H23CX653 GDLs. As can be seen from Figure 3 and Figure 4, the maximum power of hot-pressing MEAs with a co-polynaphthoyleneimide membrane only slightly increases when H23CX653 GDLs developed for a fuel cell operation under high humidity conditions were used instead of standard H23C3 GDLs.

The maximum power values for MEAs with the standard H23C3 GDL and H23CX653 GDL were 365 mW/cm^2^ and 383 mW/cm^2^, and the operating ones (at an output cell voltage of 0.6 V) were 309 mW/cm^2^ and 322 mW/cm^2^, respectively. However, as expected, the use of the H23CX653 GDL instead of the H23C3 GDL resulted in a significant reduction in electrode flooding. It is clearly seen that *I*–*U* dependence becomes more linear at high current densities for MEAs with H23CX653 GDLs (Figure 4) in comparison with MEAs using H23C3 GDLs (Figure 3). In addition, this GDL replacement allowed us to reduce the activation losses from 0.14 V for the MEA with the standard H23C3 GDL to 0.06 V for that with the H23CX653 GDL.

The next stage of this work involved reducing the ohmic interface resistance. For this purpose, MEA fabrication pressure was varied to determine its optimal magnitude. Figure 5a shows the *I*–*U* and *I*–*P* curves of MEAs that were prepared at hot-pressuring parameters of 3 MPa, 8 MPa, 13 MPa, and 18 MPa and a temperature of 130 °C in all cases. As follows from the data presented in Figure 5a, the drop of the voltage due to the activation losses near *U*_OCV_ of all studied MEAs is ~0.06 V. This indicates the correctness of the calculations of the platinum loading and gas flows, when the reaction rate on the electrodes is practically the same regardless of the interface contact.

As can be seen from the *I*–*U* curves shown in Figure 5a, the ohmic losses contribution of MEAs fabricated at a pressure of 3 MPa is significantly higher than those fabricated at 8 MPa. This indicates an insufficient interface contact between the catalyst layers and the membrane. The maximum and operating (at an output voltage of 0.6 V) powers of the MEA based on the co-PNIS_70/30_ membrane fabricated at a pressure of 3 MPa proved to be only 250 mW/cm^2^ and 223 mW/cm^2^, respectively (Figure 5b). Thus, the obtained power values are ~40% lower than, for example, for the MEA fabricated at 8 MPa. However, the open circuit potential of MEAs fabricated at a pressure of 3 MPa is the highest (U_OCV_ ≈ 0.96 V) among other studied fuel cells (Figure 5a) since at this pressure the membrane is deformed to the least extent, resulting in the minimal crossover value.

The maximum and operating powers continuously increase with the application of higher pressure (up to 13 MPa) during MEA fabrication and reach values of 438 mW/cm^2^, and 403 mW/cm^2^, respectively (Figure 5b). Nevertheless, in this case, partial electrode flooding is observed at the current densities of more than 1000 mA/cm^2^. This effect is caused by a decrease in the GDL porous channel diameter during a hot-pressing procedure, which leads to a decrease in the gas diffusion efficiency. However, it is important to note that the degree of flooding, in this case, is significantly less than when using the H23C3 GDL, which is developed for fuel cell operations under low humidity conditions (Figure 5a). Moreover, the flooding effect is observed only at a voltage of less than 0.4 V, which is lower than the operating potential (0.6 V) and can be considered uncritical.

A further increase in MEA fabrication pressure to 18 MPa results in a significant decrease in the power characteristics. In this case, the maximum power is only 274 mW/cm^2^, and the operating power (cell output voltage of 0.6 V) is 254 mW/cm^2^. This drop in power, as compared with fabrication pressure at 13 MPa, occurs due to several reasons. Firstly, the membrane becomes more gas-permeable with an increase in mechanical loads, which is also evidenced by a decrease in the open circuit voltage value from 0.96 V to 0.87 V with a pressure increase from 13 MPa to 18 MPa (Figure 5a). Secondly, the GDL pores are strongly deformed at high pressures, which leads to a significant narrowing of the gas transport channels. As a result, the GDL critical flooding is even observed at current densities of 570 mW/cm^2^ at the MEA fabrication pressure of 18 MPa. Thus, it can be concluded that the pressure of 18 MPa is strongly exceeded for the MEA fabrication with high performance.

Summarizing the obtained data, it can be concluded that the pressure range for the fabrication of high-performance MEAs based on a co-polynaphthoyleneimide membrane with the H23CX653 GDL should be insight from 8 MPa to 13 MPa. It is impossible to achieve a satisfactory interface contact between the electrodes and the membrane at pressure *p* < 8 MPa, whereas a partial collapse of the GDL pores occurs at higher pressure *p* > 13 MPa, resulting in electrode flooding. Therefore, for the final research, MEAs were prepared at 10 MPa.

The next stage in the development of MEA fabrication technology based on a co-polynaphthoyleneimide membrane was the determination of the operating temperature at which the fuel cell reveals more efficient performance. The MEA current–voltage and power characteristics were analyzed at 30 °C, 45 °C, 60 °C, 65 °C, 70 °C, and 80 °C (Figure 6a). All MEAs for these measurements were fabricated at an optimal pressure of 10 MPa.

As can be seen from the data presented in Figure 6a,b, the efficiency of the MEA based on a co-PNIS_70/30_ membrane increases with the temperature increase from 30 °C to 65 °C. The maximum power values at temperatures of 30 °C, 45 °C, 60 °C, and 65 °C are 253 mW/cm^2^, 346 mW/cm^2^, 497 mW/cm^2^, and 535 mW/cm^2^, and the operating power (at 0.6 V) is 245 mW/cm^2^, 295 mW/cm^2^, 374 mW/cm^2^, and 415 mW/cm^2^, respectively. However, it is found that the power characteristics decrease with a further increase in the temperature. For example, the maximum MEA power at temperatures of 70 °C and 80 °C are already 474 mW/cm^2^ and 383 mW/cm^2^, and the operating powers (at 0.6 V) are 360 mW/cm^2^ and 322 mW/cm^2^, respectively. A similar tendency towards a decrease in the MEA power characteristics based on ion-crosslinked polymers was also observed by other researchers [48]. The remarkable power output decrease at ~80 °C is associated with a decrease in the stability of zirconium ion cross-link, connecting single polymer blocks (Figure 1b). This leads to a decrease in the proton conductivity of membranes based on such polymers. However, it is important to note that this does not affect the chemical stability of the polymer itself but determines the temperature range for more efficient proton transport. As can be seen from the dependencies presented in Figure 6a, the highest level of the ohmic losses of the MEA based on the co-PNIS_70/30_ membrane is observed at 30 °C. This may be due to the fact that at low temperatures, there is a large interface resistance contribution between the catalyst layer and the co-polynaphthoyleneimide membrane. This feature of fuel cells based on hydrocarbon membranes is described in [49,50].

Thus, the main contribution to the MEA ohmic losses in the studied temperature range results from the following factors:At temperatures of T > 70–80 °C, fuel cell ohmic resistance increases due to a decrease in the stability of the Zr cross-link;At temperatures of T < 55–60 °C, an increase in the fuel cell ohmic resistance is mainly associated with an increase in the interface resistance between the catalytic layer deposited onto the GDL and the membrane.

Hence, an MEA based on a co-polynaphthoyleneimide membrane operates most effectively at 60–65 °C.

### 3.3. Comparison of the Performance of an MEA Based on a Co-Polynaphthoyleneimide Membrane Fabricated by a Modified Technology and an MEA Based on a Nafion Membrane Fabricated by a Standard Technology

For the final studies, the MEA based on the co-PNIS_70/30_ membrane was prepared by pressing the H23CX653 GDL and a catalyst layer against the membrane at 130 °C and at a pressure of 10 MPa for 3 min. Figure 7 presents the current–voltage and power characteristics of this MEA in comparison with similar characteristics of the MEA based on Nafion 212 membrane fabricated according to the standard technology. The measurements were carried out at 65 °C and 100% humidity of the supplied fuel and air (oxidizer).

It can be seen from Figure 7 that the operating powers (at an output cell voltage of 0.6 V) of MEAs based on co-PNIS_70/30_ and Nafion 212 membranes almost coincide and are 415 mW/cm^2^ and 419 mW/cm^2^, respectively. However, the maximum power of the MEA based on the Nafion 212 membrane is ~20% higher (674 mW/cm^2^) than that of the MEA based on the co-PNIS_70/30_ membrane (535 mW/cm^2^). Our analysis of the *I*–*U* curves clearly indicates that the main restrictive factor for reaching high power is a slightly higher level of the ohmic losses of the MEA based on a co-polynaphthoyleneimide membrane as compared with the level of the ohmic losses of the MEA based on a perfluorinated membrane. The difference in the protonic conductivity of these membranes is the same (20% [32]) as the difference in the maximum power.

Thus, the technology of the fabrication of the MEA based on a co-polynaphthoyleneimide membrane developed in this study, apparently, reached the maximum power indicators. A slight difference in the operating power (less than 1%) and a difference in the maximum output power of ~20% allows us to conclude that co-polynaphthoyleneimide membranes are very promising as an alternative to fluorinated analogs for their commercial application in a hydrogen-air fuel cell.

## 4. Conclusions

The following conclusions can be drawn from the results presented in this work. To fabricate high-performance MEAs based on a co-polynaphthoyleneimide membrane, an H23CX653 GDL developed for fuel cell operation under high humidity conditions should be used. This is necessary because of the higher moisture absorption of this type of membrane compared to the perfluorinated analogs. The H23CX653 GDL allows one to remove moisture more efficiently when the fuel cell operates under increased hydration conditions, which also prolongs its durability. It is also important to carry out the hot-pressing procedure of the GDL with a catalyst layer on the membrane in the pressure range from 8 to 13 MPa at 130 °C. The hot-pressing parameters lead to the creation of an optimal interface ohmic contact and do not disturb the connectivity of the channels for gas transport in the porous GDL medium.

The technology described in this article allows one to create single fuel cells (MEA) based on co-polynaphthoyleneimide membranes with high-power characteristics. It is found that the optimal operating temperature of a fuel cell based on a co-PNIS_70/30_ membrane is in the range of 60–65 °C. At the same time, the maximum power of the MEA fabricated according to the developed technology is ~535 mW/cm^2^, and the operating power (at 0.6 V) is 415 mW/cm^2^. A comparison with similar characteristics of the MEAs based on the commercial Nafion 212 membrane shows that the operating powers are almost the same (the difference is less than 1%), and the maximum output power of the MEA on the co-PNIS_70/30_ membrane is only ~20% lower. It is concluded that the developed technology could help with the fabrication of competitive fuel cells based on a fluorine-free and cost-effective membrane.

## Figures and Tables

**Figure 1 membranes-13-00485-f001:**
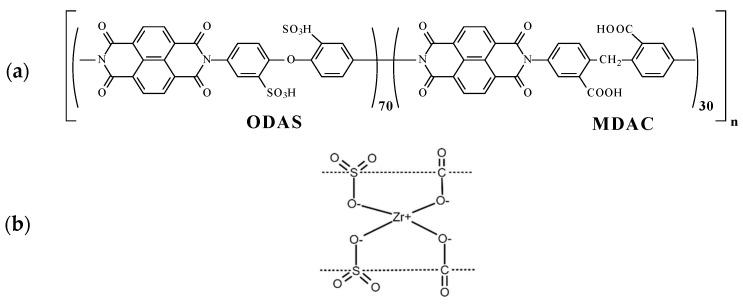
(**a**) The chemical structure (polymer elementary unit) of sulfonated co-polynaphthoyleneimide (co-PNIS_70/30_) polymer and (**b**) the principle of zirconium cross-linking of polymer elementary units between ODAS block (left) and MDAC block (right).

**Figure 2 membranes-13-00485-f002:**
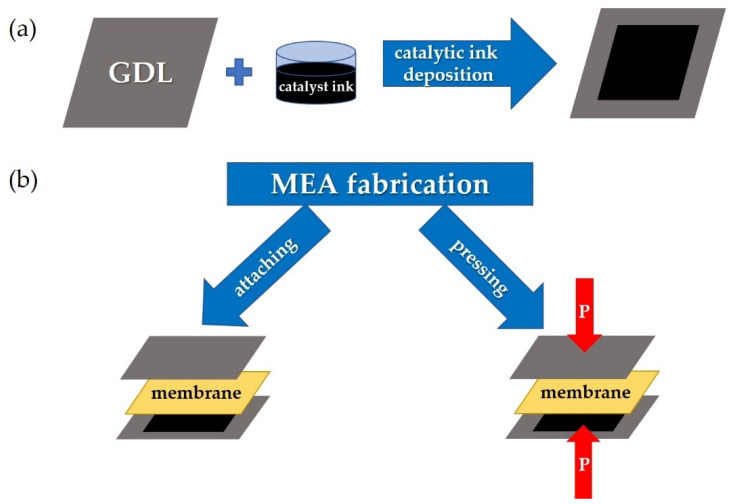
The step sequences for a gas diffusion electrode preparation based on GDL with a deposited catalyst (**a**) and MEA fabrication (**b**) by means of attaching electrodes to the membrane (left arrow) and by means of hot-pressing the membrane with the electrodes (right arrow).

**Figure 3 membranes-13-00485-f003:**
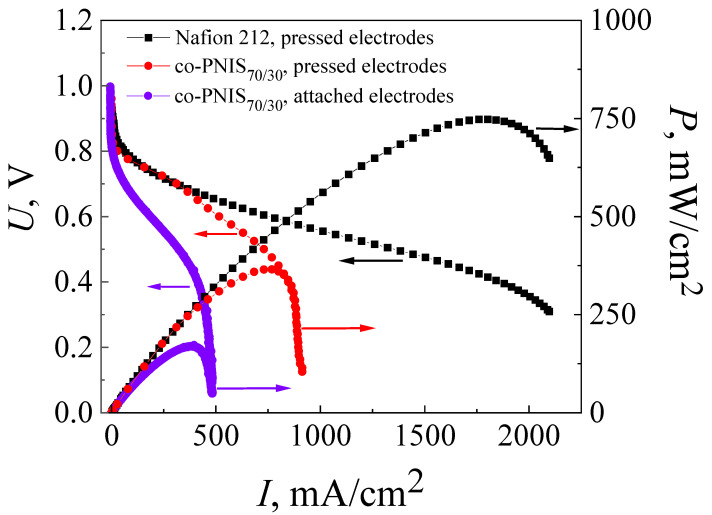
Current–voltage (*I*–*U*) and power versus current (*I*–*P*) characteristics of MEAs based on a co-polynaphthoyleneimide membrane, where violet curves belong to the attached electrodes and red curves belong to the hot-pressed electrodes fabricated using the standard technology. These characteristics are also given for MEA fabricated by the standard hot-pressing technology with Nafion 212 membrane (black curve). The studies were carried out at a temperature of 80 °C and 100% humidity of the supplied gases (H_2_ and air).

**Figure 4 membranes-13-00485-f004:**
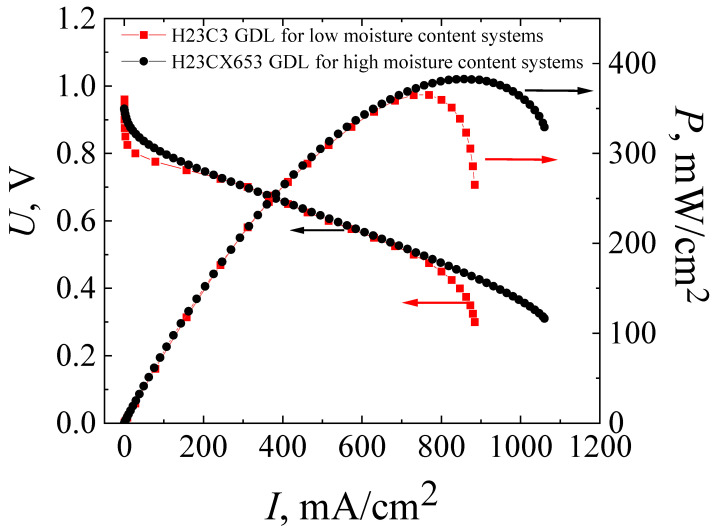
Current–voltage (*I*–*U*) and power (*I*–*P*) characteristics of MEAs based on a co-polynaphthoyleneimide membrane with H23C3 GDL (red curves) and with H23CX653 GDL (black curves), which were fabricated using the standard hot-pressing technology. The studies were carried out at a temperature of 80 °C and 100% humidity of the supplied gases (H_2_ and air).

**Figure 5 membranes-13-00485-f005:**
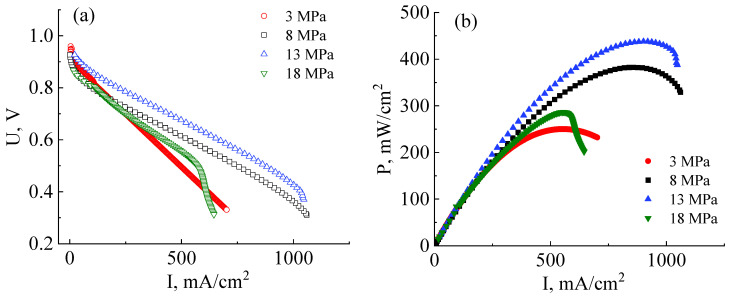
Current–voltage (**a**) and power (**b**) characteristics of MEAs based on a co-polynaphthoyleneimide membrane prepared at different pressures and investigated at a temperature of 80 °C and 100% humidity of the supplied gases (H_2_ and air).

**Figure 6 membranes-13-00485-f006:**
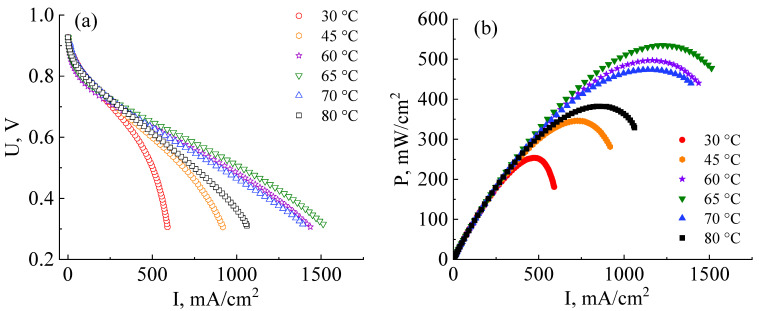
Current–voltage (**a**) and power (**b**) characteristics of MEAs based on a co-polynaphthoyleneimide membrane at the different operating temperatures and 100% humidity of the supplied gases (H_2_ and air). All MEAs were fabricated at a pressure of 10 MPa.

**Figure 7 membranes-13-00485-f007:**
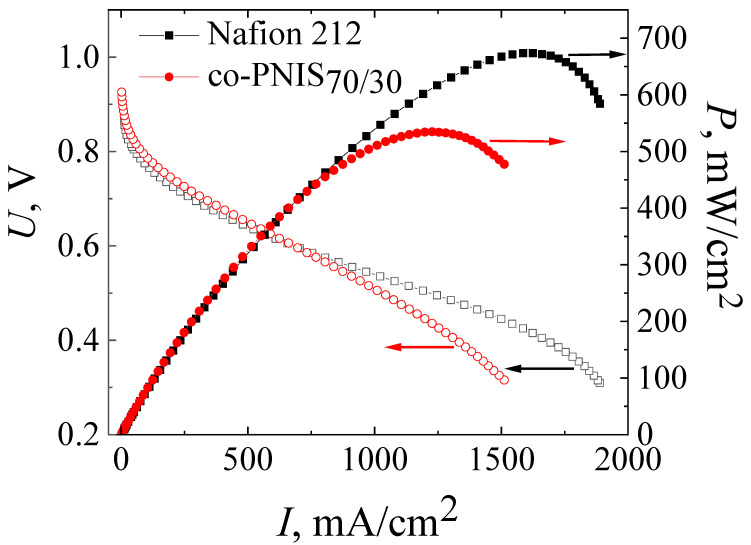
Current–voltage and power characteristics of MEAs based on co-PNIS_70/30_ membrane fabricated by a modified technology (red curves), and that based on Nafion 212 membrane fabricated by a standard technology (black curves). The studies were carried out at 65 °C and 100% humidity of the supplied gases (H_2_ and air).

## Data Availability

Not applicable.

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
