# Peer review of "Development of High-Performance Hydrogen-Air Fuel Cell with Flourine-Free Sulfonated Co-Polynaphthoyleneimide Membrane†"

_membranes, 2023, doi:10.3390/membranes13050485_

Round 1

Reviewer 1 Report

Recently, high-temperature proton exchange membrane fuel cells have been studied because of their advantages in heat rejection. However, this paper reports that the invented membranes have a stability problem even at 80oC. And I couldn’t find valuable data without the optimization of the MEA conditions. The effort for optimizing the MEA performance is interesting and the explanation of it seems to be reasonable. I recommend that the authors should prove their new membrane’s novelty and values. This paper can be published after the major revision:

1. Please explain the novelty and difference of the membrane compared with other membranes (ex. Nafion 212).

2. Please inform the active area. If the area is 5 cm2, the flow rate of air(2500 ccm) is too high to observe mass-transfer limitation. The flow rate of air should be less than 500 ccm. Please suggest the results of them.

3. Please provide the EIS data. It should be much helpful to understand the HFR, charge transfer resistance, and mass-transfer resistance. I believe these data would make your paper more abundant.

4. Please measure the contact angles of bare GDLs and catalyst layer/GDLs. It might suggest the hydrophobicity of the GDLs, preventing flooding.

5. The authors said OCVs are strongly related to the hydrogen crossover, but the evidence was not enough. Please provide the results of the hydrogen crossover current.

5. The authors are insisting the MEA operation with temperature over 70-80oC makes the problem of Zr-cross stability. Do you think Zr-cross stability could be recoverable? Have you done the MEA test at 65oC after the MEA test at 80oC? If the performance at 65oC after the MEA test at 80oC could be similar to the original performance, it could be strong evidence for the recovery of Zr-cross.

Reviewer 2 Report

The mauscripot reports on an interesting membrane for PEM-FCs. It needs extensive revision befor publication, for some suggestions see the annotated manuscript.

Author Response

Dear reviewer!

Thank you very much for your review.

Reviewer 3 Report

The manuscript reported the development of the PEMFCs with fluorine-free sulfonated co-polynaphtoyleneimide (co-PNIS) membranes. It is found that the optimal operating temperature of a PEMFC based on a co-PNIS membrane with hydrophilic/hydrophobic ratio 70/30 is 60-65 °C. A comparison with similar characteristics of the MEA based on Nafion 212 shows that the values of operating performance are almost the same. It can be concluded that the developed technology allows to create competitive PEMFCs with a fluorine-free, cost- effective co-PNIS membrane.

I consider the content of this manuscript will definitely meet the reading interests of the readers of the Membranes journal. However, there are certain English spelling and grammar issues, and also the discussion and explanation should be further improved. 

I suggest giving a minor revision and the authors need to clarify some issues or supply some more experimental data to enrich the content. This could be comprehensive and meaningful work after revision.

Detailed comments can be found in the PDF file.

Round 2

Reviewer 1 Report

The authors have responded to my questions and suggestion properly. The authors emphasized their membrane’s value and potential through the additional explanation. Even though the membranes were not better than commercial Nafion membrane, as the author said, it has a high potential for improvement and a low cost for making the film. By the way, I have a question/suggestion about the Zr cross. It seems to be an interesting result that the Zr cross phenomenon is recoverable. If you find the reason what the mechanism makes the performance low, I think it could be an interesting study of a solution for them. Hope the research about them could be published soon. This paper can be published in Membranes.